# Surface-Enhanced Raman Spectroscopy (SERS) Study Using Oblique Angle Deposition of Ag Using Different Substrates

**DOI:** 10.3390/ma12101581

**Published:** 2019-05-14

**Authors:** Jaeyeong Lee, Kyungchan Min, Youngho Kim, Hak Ki Yu

**Affiliations:** 1Department of Energy Systems Research, Ajou University, Suwon 16499, Korea; smuff20@ajou.ac.kr (J.L.); haist169@ajou.ac.kr (K.M.); kimyh9347@naver.com (Y.K.); 2Department of Materials Science and Engineering, Ajou University, Suwon 16499, Korea

**Keywords:** oblique deposition, SERS, Ag nanogap

## Abstract

The oblique angle deposition of Ag with different deposition rates and substrates was studied for surface-enhanced Raman spectroscopy (SERS) efficiency. The deposition rate for the Ag substrate with maximum SERS efficiency was optimized to 2.4 Å/s. We also analyzed the morphology of Ag nanorods deposited at the same rate on various substrates and compared their SERS intensities. Ag deposited on SiO_2_, sapphire, and tungsten showed straight nanorods shape and showed relatively high SERS efficiency. However, Ag deposited on graphene or plasma-treated SiO_2_ substrate was slightly or more aggregated (due to high surface energy) and showed low SERS efficiency.

## 1. Introduction

Surface-enhanced Raman spectroscopy (SERS) has been actively studied due to its ability to detect the traces of target molecules [1,2]. The nanostructures of materials with the SERS effect are synthesized and used as substrates to detect the target analyte [3,4]. It is known that the SERS effect is caused by the following two effects. The first is the electromagnetic effect, localized surface plasmon resonance (LSPR) [5,6], which refers to the amplification of the analyte signal by plasmon resonance on the surface of the nanostructured material (especially the metal). The second is the chemical effect [7,8]. Amplification of the signal occurs by charge transfer between the SERS material and the analyte. The SERS effects of organic and inorganic semiconductor materials are largely dependent on the chemical effects.

So far, noble metals such as Ag [9,10], Au [11,12], Cu [13], and inorganic semiconductor materials such as ZnO [7], TiO_2_ [8], and organic semiconductors [14] have been studied as materials of SERS substrates. Among them, Ag has been most actively studied due to its wide reactivity in the visible range [15] and relatively low cost compared to Au and Pt, and relatively high stability from corrosion compared to Cu [16]. The shape and spacing of the nanostructures, as well as the materials, are very important for the SERS substrate. Sputtering [17,18], solution method [19,20], and electrochemical reaction method [21] were studied to control them. The solution method is advantageous for synthesizing nanostructured SERS substrate, but it takes longer than other methods. Lithography-based sputtering is convenient for fabricating nanostructured SERS substrate, but it requires a lot of money and time. In recent years, research has been reported on obtaining a nanorod-structured SERS substrate by simply tilting the substrate during sputtering [22,23,24]. In this way, SERS substrates can be fabricated at a lower cost in a shorter time than other methods.

In this study, we fabricated Ag nanorod arrays for SERS substrates using this oblique angle deposition method and investigated the effects of various deposition rates and substrates of them. The difference in the surface energy depending on the substrate changes the morphology of the Ag film. Basically, the deposition of Ag by oblique angle deposition first nucleates the Ag on the substrate, which causes the shadow effect and makes Ag grow into nanorods [25]. However, if the surface energy is very small or large, Ag will grow to other morphology because the size of Ag changes during nucleation. The fabricated Ag nanorods arrays were confirmed by a scanning electron microscope (SEM) and the SERS efficiency of each substrate was analyzed using Rhodamine 6G (R6G) as a target molecule.

## 2. Materials and Methods

### 2.1. Oblique Angle Deposition of Ag

Ag nanorod arrays were deposited by oblique angle deposition using an electron beam evaporator at room temperature. The deposition angle was 88°. Figure 1 shows the schematic image of the oblique angle deposition. The deposition rates were 1 Å/s, 1.7 Å/s, 2.4 Å/s, and 3.1 Å/s, and the base pressure was below 5 × 10^−6^ Torr. The deposition times were 83 min, 49 min, 34 min, and 26 min, and the final thickness was 175.32 nm, 197.40 nm, 222.08 nm, and 285.71 nm, respectively. Various substrates were used to observe the tendency of Ag deposition according to the surface energy of the substrate. Si/SiO_2_ which is most commonly used as a substrate and graphene (G) with lower surface energy were selected. Tungsten (W), sapphire, and O_2_ plasma-treated Si/SiO_2_(P-Si/SiO_2_) were used as substrates with high surface energy (high wettability). Table 1 summarizes all the samples produced. The Si/SiO_2_ and sapphire substrates were rinsed sequentially with acetone, isopropyl alcohol, and deionized water for 3, 3, and 2 min prior to deposition. The tungsten layer was deposited on rinsed Si/SiO_2_ with magnetron sputtering. The sputtering was performed at 100 W, 20 sccm of Ar flow, 10^−6^ Torr of base pressure, and at room temperature. For the G substrate, the graphene grown on Cu foil through chemical vapor deposition was transferred to the Si/SiO_2_ substrate as written in our former articles [26]. Finally, for the P-Si/SiO_2_ substrate (and O_2_ plasma etched sapphire), each wafer was treated with reactive-ion plasma with 50 W, and 50 sccm O_2_ flow for 5 min.

### 2.2. Characterization

The Ag nanorods arrays deposited by oblique angle deposition were characterized by SEM, and XRD with respect to the deposition rates and the substrates. The morphology of the Ag nanorods arrays was observed by the FE-SEM (Hitachi S-4800) with top view and a cross-sectional view, and XRD patterns were obtained with Rigaku MiniFlex spectrometer (3 kW, Cu-Kα, HD307172). The contact angle measurement was conducted with 100 µL of pure water drop and calculated with a half-angle method. The SERS measurement was carried out with 100 ppm (2.1 × 10^−4^ M) R6G solution and the Raman spectra were obtained by high-resolution Raman spectrometer (HORIBA, Jobin Yvon, LabRam HR Evolution) with 633 nm He-Ne laser as the excitation source, 1% of laser power, and the illuminating spot size was about 0.89 μm in a diameter through a 100× objective. The acquisition time is 1 s with 2 accumulation times. Five spectra of each sample were obtained and averaged.

## 3. Results

### 3.1. Differences in Ag Nanorods According to Deposition Rate

Differences of the Ag nanorods arrays for different deposition rates were analyzed, shown in Figure 2. SEM images in Figure 2a show the morphology of deposited Ag layer. When deposited normally (without tilting), the Ag film showed a flat surface as shown in Figure 2a (1), while the Ag layers deposited with oblique angle deposition grew as nanorod-like structures. The tilted angles of the Ag nanorods shown in Figure 2a (2–5) were 23°, 26°, 33°, and 38°, respectively. As the deposition rate increases, the tilted angle of the Ag nanorods increase. When the deposition rates were 1 Å/s and 1.7 Å/s, it can be confirmed that the rods are connected to the other rods. When the deposition rate was higher than 2.4 Å/s, the rods were not connected but stood alone. However, when the deposition rate became 3.1 Å/s, the number of rods per unit area increased and became relatively dense. The densities of the nanorods at 2.4 Å/s and 3.1 Å/s are 1.28 × 10^−4^ nm^−2^ and 2.8 × 10^−4^ nm^−2^, respectively. As shown in Figure 2b, SERS spectra of R6G on the Ag substrates which are deposited with each rate were observed. Peaks corresponding to the spectrum of R6G were observed, which are shown in Table 2. For the quantitative analysis, the SERS spectra of R6G adsorbed on the normally deposited Ag was obtained and the SERS intensities of the oblique angle-deposited Ag at 612 cm^−1^ were divided by the spectrum of it. As shown in Figure 2c, the SERS efficiency of Ag substrate with the deposition rate of 1 Å/s and 1.7 Å/s (compared to the normally deposited Ag) were about 300 times. When the deposition rate was 2.4 Å/s, the SERS efficiency increased to about 850 times. It is reasonable to assume that R6G can be detected at lower concentrations because the intensity of R6G decreases linearly as R6G concentration decreases [27]. However, when the deposition rate increased to 3.1 Å/s, the SERS efficiency is significantly reduced to about 70. At lower deposition rates, the light of the R6G molecule is scattered by the LSPR effect of one Ag rod that the R6G molecule was absorbed on because of the relatively large spacing between Ag nanorods and the low surface area due to the connection of rods, When the spacing between nanorods narrows to a certain interval, a nanogap (called hot spot) is created in which the R6G molecule is affected by several nanorods. Therefore, the highest SERS efficiency was obtained when the deposition rate was 2.4 Å/s. Finally, at 3.1 Å/s, the density of nanorods is so high that the surface area that R6G can be absorbed on reduced, resulting in a reduction in SERS efficiency.

### 3.2. Differences in Ag Nanorods According to Substrate

The growth characteristics of Ag nanorods on each substrate were observed. Si/SiO_2_, Sapphire, W, G, and P-Si/SiO_2_ were used for the substrates. In Figure 3a, SEM images of Ag nanorods arrays for each substrate are shown. In the case of Si/SiO_2_ and W, Ag nanorods of similar shape were observed. On the sapphire substrate, Ag was deposited like film at first and then grew like nanorods similar to Si/SiO_2_ and W. In the case of Ag deposited on G, several nanorods were connected in one line and seem to have a wave-like structure rather than nanorod arrays. Ag deposited on P-Si/SiO_2_ was clumped to such an extent that it was difficult to identify the shape of the rod. The SERS intensities for each substrate are shown in Figure 3b, and the relative SERS efficiencies are plotted in Figure 3c, the same way as in Figure 2c. Ag nanorods on Si/SiO_2_, sapphire, and W which had similar shapes showed similar SERS efficiencies. In the case of Ag nanorods substrate deposited on G, SERS efficiency decreased due to surface area decrease due to rod clustering. The Ag nanorods deposited on P-Si/SiO_2_ showed the lowest SERS efficiency, which is because of the large decrease of the surface area that the R6G can be affected. This difference in efficiency can be explained by assuming that the surface of P-Si/SiO_2_ is flat. If the same degree of surface plasmon resonance occurs on all surfaces of Ag, the increase in SERS effect can be explained by an increase in the surface area of Ag. When the average diameter and length of Ag nanorods exposed to light are 100 nm and 200 nm, the surface area per Ag nanorod is 20,000π × 10^−6^ μm^2^, and the number of nanorods per 1 μm^2^ is 1.28 × 10^2^. The surface area corresponding to 1 μm^2^ in flat Ag surface is 8.042 μm^2^ in Ag nanorods arrays. This 8-fold increase in surface area corresponds to an increase in SERS efficiency of about 8 times as shown in Figure 3c This can also be explained by the UV-vis spectra of Ag deposited on the sapphire and O_2_ plasma etched sapphire shown in Figure 4. Ag on O_2_ plasma etched sapphire has lower absorbance of light corresponding to 633 nm. Less absorption of light indicates a lower degree of surface plasmon resonance.

### 3.3. Contact Angle Differences According to Substrate

The contact angle of each substrate was measured to find out the cause of the change in growth characteristics of the Ag rods according to the substrate. Figure 5a is the contact angle images of water droplets on each substrate. The contact angle was 46° for Si/SiO_2_, 24° for sapphire, 12° for tungsten, 61° for graphene and 18° for P-Si/SiO_2_. Figure 5b are SEM images of Ag deposited in a thickness of 30 nm with oblique angle-deposition. For Si/SiO_2_, Ag was deposited like a nanodot. For W, although it showed the lowest contact angle, Ag was deposited in a nanodot shape similar to that of Si/SiO_2_. This is because when Ag nuclei are formed on W, Ag atoms are preferentially adsorbed on Ag nuclei rather than W surface [32]. In sapphire and P-Si/SiO_2_ which have lower contact angles than Si/SiO_2_, Ag was deposited as wide as a film. In the case of sapphire, since a complete Ag film is formed, when Ag is deposited thereon, Ag nuclei are formed and undergo nanorod-like growth when deposited on a new substrate. On the other hand, on P-Si/SiO_2_, which has a high coverage but not a complete film, the adsorption of Ag occurs only on the Ag islands because the shadow regions are formed due to the already deposited Ag islands. Since the coverage of these islands is very high at this time, Ag is eventually deposited closer to the film than the nanorods arrays. In the case of G, which has the highest contact angle, Ag islands are deposited in very low coverage. Because of the low wettability of graphene, Ag nuclei are not produced much, and growth of the generated nuclei mainly occurs. The nuclei grow to become islands and Ag deposited on the island eventually form a wave-like structure.

## 4. Conclusions

In this study, Ag nano-columns were deposited with electron beam evaporator using oblique angle deposition. The optimized deposition rate was 2.4 Å/s, which showed a SERS efficiency of about 850 times compared to a normally deposited Ag film. When the difference in Ag growth between the various substrates was compared, it was observed that when the wettability was very low (P/Si/SiO_2_) or very high (G), Ag did not grow like nanorods but grew like clumpy structure, while it deposited like nanorods on the other substrates (Si/SiO_2_, sapphire, W). They showed low SERS efficiency compared to nanorod arrays due to the low surface area.

## Figures and Tables

**Figure 1 materials-12-01581-f001:**
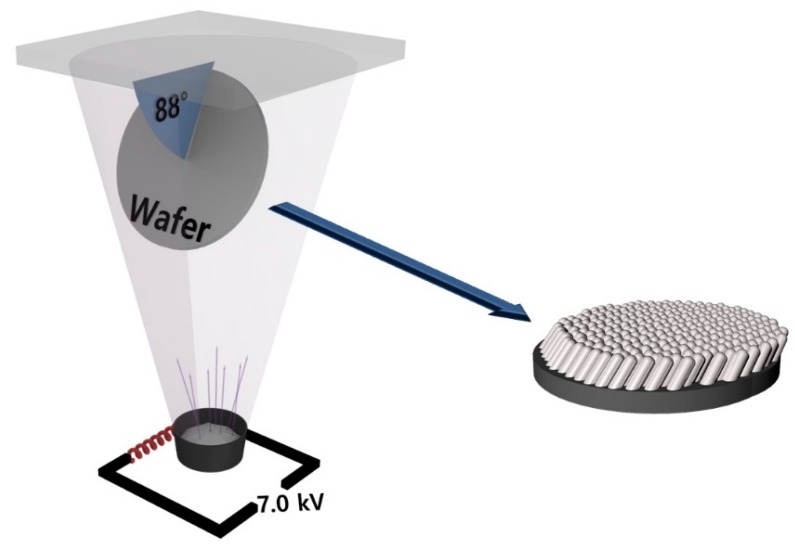
Schematic image of oblique angle deposition with electron beam evaporator.

**Figure 2 materials-12-01581-f002:**
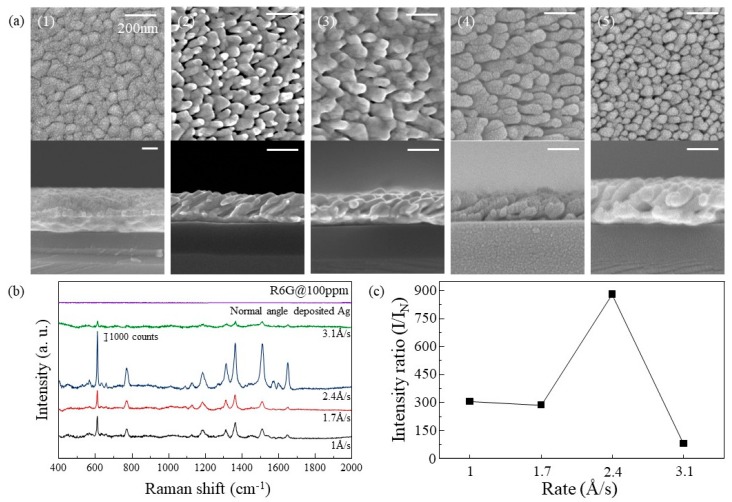
(**a**) Top view(**upper**) and cross-section(**below**) SEM images of (1) normally deposited Ag film and (2–5) Ag nanorods arrays deposited at different rates of 1, 1.7, 2.4, and 3.1 Å/s, respectively. Si/SiO_2_ was used for the substrate. (**b**) SERS spectra of R6G absorbed on Ag in (**a**). (**c**) SERS efficiency of each Ag substrate.

**Figure 3 materials-12-01581-f003:**
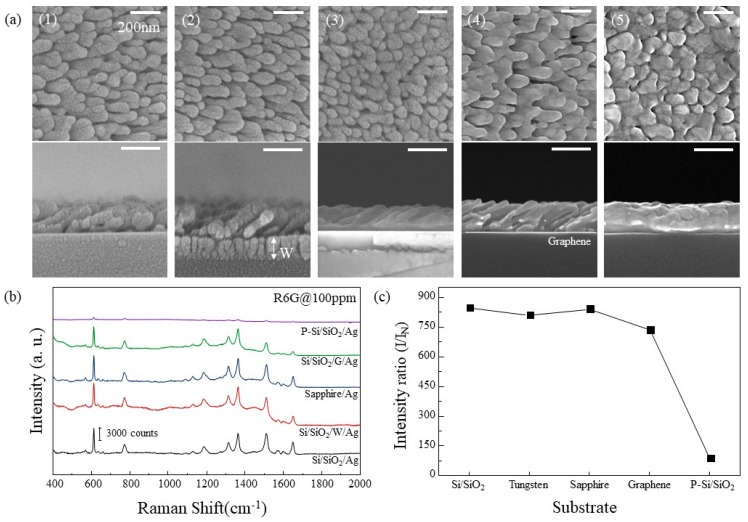
(**a**) Top view(**upper**) and cross-section(**below**) SEM images of Ag nanorods arrays deposited on different substrates at 2.4 Å/s of deposition rate. (**1**) Si/SiO_2_, (**2**) Si/SiO_2_/W, (**3**) Sapphire, (**4**) Si/SiO_2_/G, and (**5**) P-Si/SiO_2_. (**b**) SERS spectra of R6G absorbed on Ag in (**a**). (**c**) SERS efficiency of each Ag substrate.

**Figure 4 materials-12-01581-f004:**
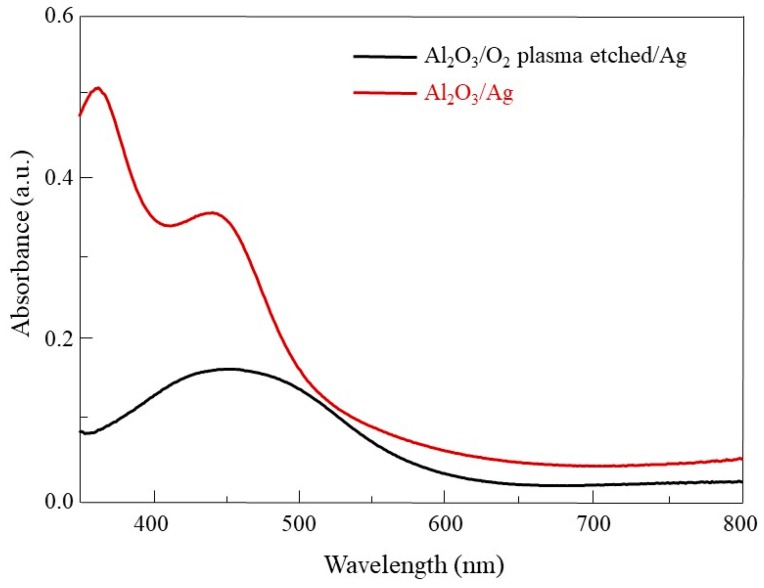
UV-vis spectra of Ag on sapphire(red) and O_2_ plasma etched sapphire(black).

**Figure 5 materials-12-01581-f005:**
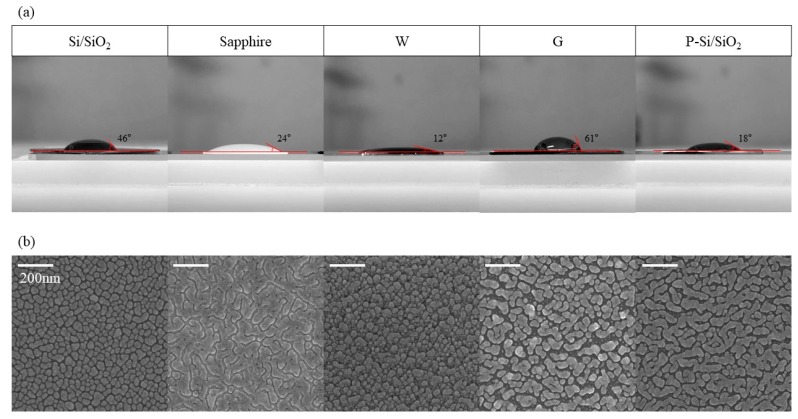
Wettability of each substrate. 30 nm of Ag was deposited at 2.4 Å/s on a different substrate. (**a**) Contact angle images of each substrate. (**b**) SEM images of 30 nm Ag deposited with oblique angle deposition on different substrates.

**Table 1 materials-12-01581-t001:** Sample treatment regarding deposition rates and substrates. Experiment 1 is an experiment on the morphology of Ag depending on the deposition rate, and Experiment 2 is an experiment on the morphology of Ag depending on the substrate. They are described in Section 3.1 and Section 3.2 respectively.

Experiment Series	Deposition Rate (Å/s)	Substrate
Experiment 1	1.0	Si/SiO_2_
	1.7	Si/SiO_2_
	2.4	Si/SiO_2_
	3.1	Si/SiO_2_
Experiment 2	2.4	Sapphire
	2.4	Si/SiO_2_/W
	2.4	Si/SiO_2_/G
	2.4	P-Si/SiO_2_

**Table 2 materials-12-01581-t002:** Assignment of Raman bands in spectra of R6G.

Raman Shift (cm^−1^)	Assignment	Reference
612	C–C–C ring in-plane bending	[28,29,30]
776	C–H out-of-plane bending	[28,29,30]
1189		[28,29,30]
1314	Aromatic C–C stretching	[28,29,30,31]
1363	Aromatic C–C stretching	[28,29,30,31]
1511	Aromatic C–C stretching	[28,29,30]
1599	Aromatic C–C stretching	[28,29,30,31]
1647	Aromatic C–C stretching	[28,29,30]

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
