# Peer review of "Surface-Enhanced Raman Spectroscopy (SERS) Study Using Oblique Angle Deposition of Ag Using Different Substrates"

_materials, 2019, doi:10.3390/ma12101581_

Round 1
Reviewer 1 Report
The authors report in this manuscript their study on the deposition of Ag nanorods by oblique e-beam deposition technique, on several different substrates and the SERS performance by adjusting the deposition rate. They gave some suggestions on the design of high performance SERS substrates by this method.
The idea is of some interests, which, however, can be observed in a vast number of publication in the literature. There are many papers already published concerning the fabrication of Ag nanorods on a variety of substrates by oblique deposition and/or glancing angle deposition by e-beam. The dependence of the SERS performance of the Ag nanorods prepared by this method, on the deposition parameters such as the substrate temperature, movement of the substrate holder, deposition angle, deposition rate, type of substrate, etc. Several models to predict the growth morphology and size of the Ag nanorods had also been proposed and were found in good agreement with the experimental observation. In my opinion, there is not enough novelty of the present study to warrant its publication in this journal.
Author Response
Thank you for the comment. We have modified by following several reviewer's comment and improved our manuscript.
Reviewer 2 Report
The manuscript (Lee et al. Surface Enhanced Raman Spectroscopy (SERS) Study using Ag Nano-Column by Oblique Deposition) reported a SERS platform produced via oblique angle deposition. In the first part of the study, the authors investigated the effect of deposition rate on the SERS sensitivity of the proposed system. They tested this system in detection of Rhodamine 6G (R6G). In the second part, the authors attempted to determine the effect of the type of the substrate (SiO2, sapphire, tungsten, graphene, and plasma-treated SiO2) on final morphology of the silver film and their relevant SERS activity. The first part is well-known and do not show any scientific novelty. However, the second part of the study is good in novelty but the data in the work is so preliminary and the manuscript was not well-written. There are many issues that must be clearified. I would recommend publication after the author address the following comments.
· The Introduction section is so preliminary. The authors must compare oblique angle deposition techniques with others (colloids, lithography) in respect of advantages and drawbacks. Also, the authors mention about how the chemical nature of the substrate affects the final morphology of the resultant metallic film.
· The title of the study is so general. For the specification, the author may consider the change of the title. The authors may also use ‘Oblique Angle Deposition’ instead of ‘Oblique Deposition’.
· Line 28-29
In addition to plasmonic and inorganic semiconductors, the organic semiconductors can be used as SERS platforms. The authors can add this information to the manuscript.
· Line 30-31
The oxidation is the major drawback of the silver compared with gold as plasmonic material. The author must reconsider this statement.
…..tively low cost, and stability from corrosion [15]……
· The position of the QCM sensor is important to determine the exact deposition rate. If the QCM sensor is not close to substrate, probably the exact deposition rate would be different from measured one. I wonder if the authors considered this one during deposition work.
· What is the deposition angle? From Figure 1, it seems that it is 98. The authors mention it in the manuscript.
· Line 73-74
………..deposition rate became 3.1 Å /s, the number of rods per unit area increased and became relatively dense…….
Please quantify number density of nanorods.
· For the quantification of SERS platforms, it would be better to calculate enhancement factors instead of compare substrate with each other.
· The Raman band assignment for R6G must be given in the manuscript.
· I could not detect any data for the surface plasmon resonance (SPR) charactersitics of the proposed SERS system. The authors must add UV-vis data to clearify thiss issue. This data also would be helful to define the SERS activity.
· There is no satisfying explanation for the origin of the SERS activity. The authors must use some SPR data or some theoretical model to determine the source of the enhancement in Raman signal. The discussion section for this part is so poor.
· There is no data in the manuscript to show the reproducibility of the proposed SERS system. The authors must collect a proper number of SERS signals from the different spots and analyze them statistically (such as relative standard deviation) to evaluate reproducibility.
· There are many grammatical errors that must be eliminated.
Author Response
Thank you for the comment, we attached as separated file.

Reviewer 3 Report
The aim of the paper is to fabricate silver films, on different types of substrates and with various deposition rates in order to test their SERS efficiency for R6G detection. The theme is appropriate for publication in this journal, but the manuscript lacks important information in presenting the results.
Strenghs of the study: the experimental techniques are appropriate for the investigations.
Weaknesses of the study: mandatory improvements of the English; no assessment of the reproducibility of the films and their SERS efficiency; it lacks important information on the experimental conditions used to fabricate the films; no practical application envisioned for the obtained films.
Specific comments:
1. Row 10. Please rephrase the first phrase from the Abstract.
2. Row 22. Please add "s" to "substrate"
3. In the phrase "Sputtering [16, 17], solution method [18, 19], and electrochemical reaction method [20] were studied to control them." ref [16] only refers to deposition of Au filmsm allthough the present manuscipt is related to Ag deposition. Please explain the connection between the indicated reference and the presented work.
4. Please rephrase the phrase from row 22-row 23.
5. Throughout the manuscipt, th authors refer to the nanostructures as "nanorods", while in the title appears as "nano-columns". Please explain.
6. The authors deposit the films using oblique angle deposition; except Figure 1 where the deposition angle appers, nowhere in the experimental section can be found the deposition angle used.
7. It is not clear how many samples were deposited. A table comprising all the samples, the substrates used, deposition rates, deposition time etc and other useful experimental details would bring a plus to the quality of the work.
8. Please explain the considerations on which the substrates were chosen.
9. Row 45-row 46. Please rephrase.
10. Why are the Tungsten and Copper intermediate layers used?
11. Material and Methods section: please give detailed information about the mangetron sputtering deposition of tungsten layer.
12. Regarding the SERS detection of R6G: a concentration of 10-4 M is pretty high for a SERS experiment. In a previous study published in 2012 (Wang et al, Appl Surf Sci 2012 https://doi.org/10.1016/j.apsusc.2012.02.129) using also a silver surface, a detection of 10-15 M concentration for R6G was possible. Please explain the choice of the 10-4 M concentration and if other lower R6G concentrations were tested using the fabricated silver films.
13. Please indicate the laser used, the exposure time, accumulations, laser power for the SERS measurements.
14. What is the angle used for the SEM measurements? PLease add it to the SEM experimental section.
15.No experimental parameters are given for the Ag deposition films. Please add to the Ag deposition section the deposition times, final thicknesses etc.
16. Figure 2. What type of substrate/s have been used?
17. Figure 3. What is/are the deposition rate/s?
18. Further information regarding the reproducibility of the Ag films should have been carried out. No information related to this is mention in the manuscript.
19. Figure 4. What is/are the deposition rate/s?
Author Response

(The authors gave the same response as above.)

Reviewer 4 Report
Some minor english language mistakes, but overall an interesting paper
It should be accepted after some minor english checks
Author Response
Thank you for the comments and we have checked English again.
Round 2
Reviewer 1 Report
The authors have made efforts to modify the manuscript. I would like to recommend acceptance of this manuscript in the revised version for publication.
Reviewer 2 Report
The revisons made by the authors are satisfying. I recommend the publication of the manuscript in its present form.
Reviewer 3 Report
The additional information given added a plus to the quality of the manuscript.